# DOA Estimation in Non-Uniform Noise Based on Subspace Maximum Likelihood Using MPSO

**Jui-Chung Hung**

Department of Computer Science, University of Taipei, Taipei 100, Taiwan; juichung@go.utaipei.edu.tw;
Tel.: +886-2-2311-3040

**Abstract:** In general, the performance of a direction of arrival (DOA) estimator may decay under a non-uniform noise and low signal-to-noise ratio (SNR) environment. In this paper, a memetic particle swarm optimization (MPSO) algorithm combined with a noise variance estimator is proposed, in order to address this issue. The MPSO incorporates re-estimation of the noise variance and iterated local search algorithms into the particle swarm optimization (PSO) algorithm, resulting in higher efficiency and a reduction in non-uniform noise effects under a low SNR. The MPSO procedure is as follows: PSO is initially utilized to evaluate the signal DOA using a subspace maximum-likelihood (SML) method. Next, the best position of the swarm to estimate the noise variance is determined and the iterated local search algorithm to reduce the non-uniform noise effect is built. The proposed method uses the SML criterion to rebuild the noise variance for the iterated local search algorithm, in order to reduce non-uniform noise effects. Simulation experiments confirm that the DOA estimation methods are valid in a high SNR environment, but in a low SNR and non-uniform noise environment, the performance becomes poor because of the confusion between noise and signal sources. The proposed method incorporates the re-estimation of noise variance and an iterated local search algorithm in the PSO. This method is effectively improved by the ability to reduce estimation deviation in low SNR and non-uniform environments.

**Keywords:** non-uniform noise; memetic algorithms; particle swarm optimization; direction of arrival estimation; subspace maximum-likelihood

## 1. Introduction

Obtaining original signal-related information from signal sources containing interference is a very important issue [1,2]. The main sources of interference in the development of mobile communication technologies are low signal-to-noise ratio (SNR) and non-uniform noise. Array signal processing technologies have been applied to estimate the direction of arrival (DOA), using sensing elements arranged in different geometries to sample the wave field and collect spatial-related information to calculate the signal source DOA [3–6]. In wireless communications, low SNR and non-uniform noise are types of propagation phenomena, which can lead to misrecognition of the signal source and significant degradation of DOA estimation performance [7–12].

Among DOA estimation techniques, the maximum-likelihood (ML) [4,5,13] and multiple signal classification (MUSIC) [14] methods are the most representative. The ML algorithm assumes that the noise has a white Gaussian distribution and that the energy is uniform. The MUSIC method uses the autocorrelation matrix of the received signal to perform feature decomposition and decomposes its feature vector into signal and noise subspaces, utilizing the characteristics of orthogonality between the signal and noise to establish the DOA search criteria of the signal source. The performance of the ML and MUSIC methods is adversely affected by low SNR, however [12]. Therefore, Ji et al. [15] proposed the spatial MUSIC algorithm, while Zhang et al. [16] used the colony algorithm to solve

computationally complex problems; however, these methods cannot deal with the DOA inaccuracy caused by excessive noise. Madurasinghe [10] suggested the power domain ML method to formulate a new objective function to solve the problem of low-energy non-uniform noise through estimation of the actual noise; however, the objective function did not propose a solution for the general low SNR and high non-uniform noise method. Wen [11] proposed the smooth space method to deal with coherent signals and non-uniform noise. This method is similar to the signal subspace projection technique under a low SNR and suffers performance loss under high non-uniform noise environments. Pesavento et al. [5] proposed a non-uniform noise and combined iterative quadratic algorithm (PIQML) to improve the estimation performance and obtain better results; however, their method shows poor performance if the non-uniform noise is too large. The judgment error will be invalid, due to early iteration. Sha et al. [17] proposed the use of projection into a subspace to establish a high-resolution estimate of the associated signal direction angle. This method can reduce the computational complexity and can handle higher resolution DOA problems, but is not suitable under low SNR conditions. This paper proposes the subspace ML (SML) method using iterated local searching by the memetic particle swarm optimization (MPSO) [8,18,19] algorithm to search the neighborhood of the signal direction, in order to build the beam-space [20]. The received data are bypassed through beamforming, which can decrease the non-uniform noise phenomenon [11,12].

The particle swarm optimization (PSO) algorithm was inspired by the social behavior of animals, such as bird flocking, swarming, and the schooling of fish. It is a branch of evolutionary algorithms, first suggested by Kennedy et al. [21]. PSO has been shown to be outstanding for the solution of DOA problems and is simple to implement [22–24]. PSO is a population-based random search optimization procedure, in which the population is called a swarm. Each swarm consists of many particles and is updated based on the influence of individual experiences, the best past experience of each individual, and the overall best experience. The swarm characteristics of parallel multi-directional search are different from the general heuristic method. The advantage of PSO is that it is simple to solve and that it has the characteristics of parallel multi-directional search, which can quickly find the optimal method but is more likely to converge to a local optimum result and does not guarantee convergence to the global optimum, especially when the objective function has a high dimension or is a complex non-linear function [25,26].

To reduce the premature convergence of PSO and to obtain an adequate solution for DOA estimation under low SNR and non-uniform noise environments, this paper proposes the combination of the iterated local search algorithm and the PSO to construct a MPSO for solving the DOA under low SNR and non-uniform noise conditions. The proposed MPSO algorithm is simple and practicable, as it adopts a first-order Taylor series expansion of the target function using the SML criterion [9,27], in order to reduce the non-uniform noise effect, therefore increasing the capacity of PSO to find the best solution. The first-order Taylor series approximates the spatial search vector and cuts it down to a direct one-dimensional optimization [20]. Simulation results show that the proposed method has a considerably improved ability to decrease the estimation bias under non-uniform noise and a low SNR environment.

The remainder of the paper is structured as follows: Section 2 describes the SML DOA estimator. Section 3 presents the SML DOA estimator using MPSO. Section 4 presents numerical simulation results, illustrating the effect of the proposed method. The final section outlines our conclusions, referring to the proposed estimator.

## 2. SML DOA Estimator

Assume a $P$ narrowband signal impinges on $M$ ($P < M$) sensors in a uniform linear array (ULA) system. The $t$th measured snapshot of the received signal is written as [1]:

$$\mathbf{x}(t) = \sum_{p=1}^{P} \mathbf{a}(\theta_p)s_p(t) + \mathbf{n}(t) = A(\boldsymbol{\theta})\mathbf{s}(t) + \mathbf{n}(t), \quad t = 1, 2, \cdots, N, \tag{1}$$

where $\theta = \begin{bmatrix} \theta_1 & \theta_2 & \cdots & \theta_P \end{bmatrix}^T$ is the unknown DOA, the superscript $T$ indicates transposition, $N$ is the number of snapshots, $\mathbf{n}(t) = \begin{bmatrix} n_1(t) & n_2(t) & \cdots & n_M(t) \end{bmatrix}^T$ is the sensor noise, $\mathbf{x}(t) = \begin{bmatrix} x_1(t) & x_2(t) & \cdots & x_M(t) \end{bmatrix}^T$, $x_i(t)$ is the $i$th sensor receiving signals, $\mathbf{s}(t) = \begin{bmatrix} s_1(t) & s_2(t) & \cdots & s_P(t) \end{bmatrix}^T$ is the $P \times 1$ vector of signal amplitudes, and $\mathbf{A}(\theta)$ is the $M \times P$ composite steering matrix, expressed as

$$\mathbf{A}(\theta) = \begin{bmatrix} \mathbf{a}(\theta_1) & \mathbf{a}(\theta_2) & \cdots & \mathbf{a}(\theta_P) \end{bmatrix}^T,$$
$$\mathbf{a}(\theta_i) = \begin{bmatrix} 1 & \exp(-j2\pi d \sin\theta_i/\chi) & \cdots & \exp(-j2\pi d(M-1)\sin\theta_i/\chi) \end{bmatrix}^T, \tag{2}$$

where $\mathbf{a}(\theta_i)$ is the steering vector, $\chi$ is the wavelength, and $d$ is the sensor spacing between two neighboring sensors. In this paper, the sensor noise, $\mathbf{n}(t)$, is considered to be non-uniform and to be a zero mean Gaussian process, such that

$$E[\mathbf{n}(t)] = 0$$
$$\mathbf{R}_n = E\left[\mathbf{n}(t)\mathbf{n}(t)^H\right] = diag\{\sigma_1^2, \sigma_2^2, \cdots, \sigma_M^2\}, \tag{3}$$

where $E[\cdot]$ is the expectation, the superscript $H$ denotes the complex conjugate transpose, $diag\{.\}$ is a diagonal matrix composed of the bracketed elements, and $\sigma_i^2$ is the $i$th sensor's noise variance. In general, the sensor noise $\mathbf{n}(t)$ is uncorrelated with all signals. The array covariance corresponding to Equation (1) can be expressed as

$$\mathbf{R} = E\{\mathbf{x}(t)\mathbf{x}^H(t)\} = \mathbf{A}(\theta)\mathbf{R}_s\mathbf{A}^H(\theta) + \mathbf{R}_n, \tag{4}$$

where $\mathbf{R}_s = E[\mathbf{s}(t)\mathbf{s}^H(t)]$ is the covariance matrix of the signal amplitudes. The array covariance matrix can be estimated by the sample average, $\hat{\mathbf{R}}$:

$$\hat{\mathbf{R}} = \frac{1}{N}\sum_{t=1}^{N}\mathbf{x}(t)\mathbf{x}^H(t). \tag{5}$$

The ML estimator for non-uniform noise can be found using the weighted least-squares approach [10], using the normalized composite steering matrix and noise component. The maximum likelihood problem becomes a least-squares solution [10]:

$$L(\theta, \sigma^2) = \min_{\theta, \sigma^2}\sum_{t=1}^{N}\left|\overline{\mathbf{x}}(t) - \overline{\mathbf{A}}(\theta)\mathbf{s}(t)\right|^2, \tag{6}$$

where $\sigma^2$ is the $M \times 1$ vector of noise variance, $|.|^2$ denotes the $l_2$ norm, $\overline{\mathbf{x}}(t)$ is the normalized receiving signal, $\overline{\mathbf{x}}(t) = \mathbf{R}_n^{-1/2}\mathbf{x}(t)$, and $\overline{\mathbf{A}}(\theta)$ is the normalized steering matrix, $\overline{\mathbf{A}}(\theta) = \mathbf{R}_n^{-1/2}\mathbf{A}(\theta)$. The $\overline{\mathbf{A}}(\theta)\mathbf{s}(t)$ of Equation (6) is separable and, for fixed $\overline{\mathbf{A}}(\theta)$, $\mathbf{s}(t)$ can be obtained by using the pseudo-inverse [9]:

$$\mathbf{s}(t) = \left[\overline{\mathbf{A}}^H(\theta)\overline{\mathbf{A}}(\theta)\right]^{-1}\overline{\mathbf{A}}^H(\theta)\overline{\mathbf{x}}(t). \tag{7}$$

Given Equation (7), substituting the ML estimator into Equation (6) results in

$$L(\theta, \sigma^2) = \min_{\theta, \sigma^2} tr\{\mathbf{P}_{\overline{\mathbf{A}}}\hat{\mathbf{R}}\}, \tag{8}$$

where $\mathbf{P}_{\overline{\mathbf{A}}} = I - \overline{\mathbf{A}}(\theta)\left[\overline{\mathbf{A}}^H(\theta)\overline{\mathbf{A}}(\theta)\right]^{-1}\overline{\mathbf{A}}^H(\theta)$, $I$ is the unit diagonal matrix, and $tr\{.\}$ is the trace of the matrix. Equation (8) is a multi-objective minimization problem. In general, $L(\theta, \sigma^2)$ is a very highly

non-linear function of $\theta$ and $\sigma^2$, and the cost function is highly non-linear; it is impossible to represent the target function with any closed-form expression [16]. However, the ML estimator still has inferior performance when the noise is non-uniform and in a low SNR environment. It is well-known that the signal subspace projection method can weaken the noise effect of the received noisy data vector [9].

Applying eigende composition, the sample covariance matrix Equation (5) can be expressed as

$$\hat{\mathbf{R}} = \sum_{m=1}^{M} \lambda_m \mathbf{e}_m \mathbf{e}_m^H \qquad m = 1, \ldots, M, \tag{9}$$

where $\lambda_1 \geq \lambda_2 \geq \cdots \geq \lambda_M$ are the eigenvalues of $\hat{\mathbf{R}}$ and $\mathbf{e}_m$ denotes the eigenvector associated with $\lambda_m$ for $m = 1, 2, \cdots, M$. The column spans of $\mathbf{E_s} = [\mathbf{e}_1, \ldots, \mathbf{e}_P]$ and $\mathbf{E}_n = [\mathbf{e}_{P+1}, \ldots, \mathbf{e}_M]$ are defined as the signal and noise subspaces, respectively. The covariance matrix can be expressed as the summation of two orthogonal components, $\mathbf{E}_s \mathbf{E}_s^H$ and $\mathbf{E}_n \mathbf{E}_n^H$. Hence, this paper adopts the signal subspace projection $\mathbf{E}_s \mathbf{E}_s^H \mathbf{x}(t)$, the projection of $\mathbf{x}(t)$ onto the columns of $\mathbf{E}_s$. Properly constructing the signal subspace projection-based approach to filter the non-uniform noise can effectively enhance the performance. The SML estimator using the ML estimator of Equation (8) can be expressed as:

$$L(\theta, \sigma^2) = \min_{\theta, \sigma^2} tr\{\mathbf{P}_{\overline{\mathbf{A}}} \mathbf{E}_s \mathbf{E}_s^H \hat{\mathbf{R}}\}. \tag{10}$$

Selection of a signal subspace under a low SNR or non-uniform noise is very difficult when using eigendecomposition. As the received signal has low SNR or non-uniform noise, $\mathbf{E}_s$ may contain a noise subspace and $\mathbf{E}_n$ may contain a signal subspace. This paper adopts the reiterated procedure of a method to reduce such confusion. First, it is assumed that the noise variance is constant for all sensors ($\lambda_M$) and that the SML estimator in Equation (10) can be expressed as

$$L(\theta) = \min_{\theta} tr\{\mathbf{P}_{\overline{\mathbf{A}}} \mathbf{E}_s \mathbf{E}_s^H \hat{\mathbf{R}}\}. \tag{11}$$

Next, using Equation (11) to obtain the DOA, the noise variance $\hat{\mathbf{R}}_n$ can be estimated by

$$\hat{\mathbf{R}}_n = \frac{1}{N} \sum_{t=1}^{N} \left| \mathbf{x}(t) - \mathbf{A}(\hat{\theta})\mathbf{s}(t) \right|^2,$$
$$\mathbf{s}(t) = \left[ \mathbf{A}^H(\hat{\theta})\mathbf{A}(\hat{\theta}) \right]^{-1} \mathbf{A}^H(\hat{\theta})\mathbf{x}(t), \tag{12}$$

where $\hat{\theta}$ is the estimated DOA. This procedure implies that, for a fixed $\hat{\mathbf{R}}_n$, the solution $\theta$ minimizes $L(\theta)$ in Equation (11), and vice versa. Once $\theta$ is obtained, a refined result for $\hat{\mathbf{R}}_n$ can be achieved using Equation (12). Hence, $\hat{\theta}$ and $\hat{\mathbf{R}}_n$ can be estimated in an iterative manner.

The steering matrix can be determined using a first-order Taylor series approximation expansion at the estimation DOA $\hat{\theta}$, expressed as [1]:

$$\overline{\mathbf{A}}(\theta) = \overline{\mathbf{A}}(\hat{\theta} + \eta\theta) \cong \overline{\mathbf{A}}(\hat{\theta}) + \eta\theta \, \overline{\mathbf{A}'}(\hat{\theta}), \tag{13}$$

where $\eta\theta$ is a small value and $\overline{\mathbf{A}'}(\hat{\theta}) = \frac{d}{d\theta} \overline{\mathbf{A}}(\theta)\big|_{\theta=\hat{\theta}}$. Substituting Equation (13) into Equation (11), the following expression can be obtained:

$$L(\theta) = \min_{\eta\theta} tr\{\mathbf{P}_{\overline{\mathbf{A}}(\hat{\theta})+\eta\theta \, \overline{\mathbf{A}'}(\hat{\theta})} \mathbf{E}_s \mathbf{E}_s^H \hat{\mathbf{R}}\}, \tag{14}$$

which is a direct one-dimensional optimization problem. Then, it can easily be shown that the optimum $\eta\theta$ is given by $\frac{d\{tr\{\mathbf{P}_{\overline{\mathbf{A}}(\hat{\theta})+\eta\theta \, \overline{\mathbf{A}'}(\hat{\theta})} \mathbf{E}_s \mathbf{E}_s^H \hat{\mathbf{R}}\}\}}{d(\eta\theta)} = 0$. The value of $|\eta\theta|$ has two characteristics [8]: first, if the value of $|\eta\theta|$ is small, it can achieve a convergence result that may be local or global. However, if the

value of $|\eta\theta|$ is large, the results may remain far from convergence. Using these characteristics of $|\eta\theta|$, $\hat{\theta}$ is updated by:

$$\hat{\theta} = \begin{cases} \hat{\theta} + \eta\theta, \; if \; |\eta\theta| > \varepsilon \\ \hat{\theta}, \; if \; |\eta\theta| \leq \varepsilon \end{cases}, \tag{15}$$

where $\varepsilon$ is the search precision value. In this paper, Equation (15) is repeated until $|\eta\theta| < \varepsilon$. The proposed procedure is as follows:

1.  Given the received signal $\mathbf{x}(t)$, compute $\hat{\mathbf{R}}$ in Equation (5) and the eigendecomposition in Equation (9).
2.  Assume that the noise variance, which is constant for all sensors, is $\lambda_M$.
3.  Estimate the DOA $\theta$ using the SML estimator in Equation (11).
4.  Update the estimate of $\theta$ in Equation (15) using the $\eta\theta$ property until $|\eta\theta| \leq \varepsilon$.
5.  Update the estimated non-uniform noise $\hat{\mathbf{R}}_n$ in Equation (12).
6.  Reiterate Steps 3 to 5 until $\hat{\theta}$ and $\hat{\mathbf{R}}_n$ converge.

From the above, we propose a new method, which is a hybrid algorithm that combines PSO and the estimated DOA $\theta$ with an iterated local search algorithm for $|\eta\theta|$.

## 3. The Proposed Method

This paper proposed a hybrid algorithm that incorporates PSO with an iterated local search algorithm. The DOA with SML estimator criterion cannot be directly carried out under a non-uniform noise and low SNR environment, in which the closed form of the contained criteria are $\{\theta, \sigma^2\}$ at the same time. Therefore, MPSO is introduced to solve the issue.

The MPSO includes the following two components: First, the PSO-based SML estimation method searches the entire space. Second, the local search using the property achieves a more accurate search around potential solutions of the first component. The design process of the MPSO is expressed below.

### 3.1. The PSO-Based SML Estimation

The PSO algorithm is an optimized search method based on a group that is easy to use, as the algorithm requires few parameters. The swarm of the PSO algorithm consists of many particles. Each individual particle represents a solution; it has its own position and velocity, the initial values of which are set randomly. Then, each particle obtains a value measure from a target function; the changing of the particle position is regulated by the value of the objective function. Our objective is to minimize the value of the fitness function. We use the fitness function $L(\theta)$ presented in Equation (11). There are three kinds of influences on the movement of particles. First, movement in the previous direction; second, movement towards the position of the individual particle's optimization situation; and, finally, movement towards the position of the global optimization situation of the overall swarm [21].

Let $S$ denote the swarm size, $\mathbf{v}_i(t) = \begin{bmatrix} v_{i,1}(t) & v_{i,2}(t) & \cdots & v_{i,P}(t) \end{bmatrix}^T$ be the current velocity, and $\theta_i(t) = \begin{bmatrix} \theta_{i,1}(t) & \theta_{i,2}(t) & \cdots & \theta_{i,P}(t) \end{bmatrix}^T$ be the current position. During each iteration, the update to the velocity $v_{i,j}(t+1)$ and position $\theta_i(t+1)$ of each particle is as follows:

$$v_{i,j}(t+1) = \kappa \cdot v_{i,j}(t) + c_1 \times \varphi_{1,i}(t) \times (p_{i,j} - \theta_{i,j}(t)) + c_2 \times \varphi_{2,i}(t) \times (g_j - \theta_{i,j}(t)),$$
$$for \; all \; i = 1, 2, \cdots, S, j = 1, 2, \cdots, P \tag{16}$$

$$\theta_i(t+1) = \theta_i(t) + \mathbf{v}_i(t+1), \tag{17}$$

where $v_{i,j}(t)$ is the velocity of the $j$th dimension of the $i$th particle, $\kappa$ is the inertia weight (this value is typically set as $0 \leq \kappa < 1$), $c_1$ and $c_2$ are set near 2.0 [27], $\varphi_{1,i}(t)$ and $\varphi_{2,i}(t)$ are uniformly distributed random numbers in the range [0,1] $p_{i,j}$ is the individual best position of the $j$th dimension of the $i$th

particle, and $g_j$ is the global best position of the *j*th dimension. The individual best position of each particle is updated using

$$\mathbf{p}_i = \begin{cases} \mathbf{p}_i, & \text{if } L(\theta_i(t+1)) \geq L(\theta_i(t)) \\ \theta_i(t+1), & \text{if } L(\theta_i(t+1)) < L(\theta_i(t)) \end{cases}, \tag{18}$$

where $\mathbf{p}_i = \begin{bmatrix} p_{i,1} & p_{i,2} & \cdots & p_{i,P} \end{bmatrix}^T$. The overall best position can be found as follows:

$$\mathbf{g} = \min L(\mathbf{p}_i), \, for \, i = 1, 2, \cdots, S, \tag{19}$$

where $\mathbf{g} = \begin{bmatrix} g_1 & g_2 & \cdots & g_P \end{bmatrix}^T$. The value of the velocity $v_{i,j}(t)$ can be limited to the range $[-v_{\max}, v_{\max}]$, in order to reduce the number of particles escaping the search space with an uncontrollable trajectory [22]. In this paper, we use $v_{\max} = 0.1 \times \theta_{\max} = 18°$ for the ULA system [8].

### 3.2. The MPSO Estimator

The MPSO proposed in this paper combines the application of a local search method into the PSO algorithm, in order to solve the problem that the SML estimator criterion cannot obtain a closed form solution. Using the PSO to estimate $\hat{\theta}$ in Equation (12), the characters of $|\eta\theta|$ in the local search method and the re-estimated $\hat{\mathbf{R}}_n$ are combined. A small $|\eta\theta|$ value will generate convergent results, while a large $|\eta\theta|$ value requires a greater time to converge and may cause a value that changes the estimated deviation towards a more correct value. The pseudocode of the Algorithm 1 MPSO is as follows:

---
**Algorithms 1. MPSO**

---

**Input**: received signal $\mathbf{x}(k)$ and set of initial values for $S, c_1, c_2, \varepsilon, \theta_{\min}, \theta_{\max}$ in MPSO.
**Output**: DOA $\theta \in \begin{bmatrix} \theta_{\min} & \theta_{\max} \end{bmatrix}$
**Step 1: Set** $t = 0$.
**Step 2: Evaluate** $\hat{\mathbf{R}}$ by Equation (5), $\lambda_M$ by Equation (9), and randomly uniformly generate $\theta_i(t)$ and $\mathbf{v}_i(t)$ for $i = 1, 2, \cdots, S$.
**Step 3: Evaluate** fitness value $L(\theta_i(t))$ in Equation (11).
**Step 4: Set** $\mathbf{p}_i \leftarrow \theta_i(t)$ and $\mathbf{g} = \min\{\mathbf{p}_i\}$.
**Step 5: Update** the velocities $v_{i,j}(t+1)$ using Equation (16) and positions $\theta_i(t+1)$ using Equation (17).
**Step 6: Evaluate** the fitness value $L(\theta_i(t+1))$ in Equation (11).
**Step 7: IF** $L(\theta_i(t+1)) < L(\theta_i(t))$, **then** $\mathbf{p}_i \leftarrow \theta_i(t+1)$.
**Step 8:** $\mathbf{g} \leftarrow \min\{\mathbf{p}_i\}$ and $\hat{\theta} \leftarrow \mathbf{g}$
**Step 9: While** $|\eta\theta| > \varepsilon$, **Do**
        **Update** $\hat{\theta} \leftarrow \hat{\theta} + \eta\theta$
        **Evaluate** $|\eta\theta|$
     **End While**.
**Step 10: Evaluate** $\hat{\mathbf{R}}_n$ in Equation (12)
**Step 11: Set** $\mathbf{g} \leftarrow \hat{\theta}$.
**Step 12: Set** $t = t + 1$.
**Step 13: Go to** Step 5 until the stopping criterion is satisfied.

---

Based on the above analysis, Figure 1 presents a flowchart of the proposed algorithm.

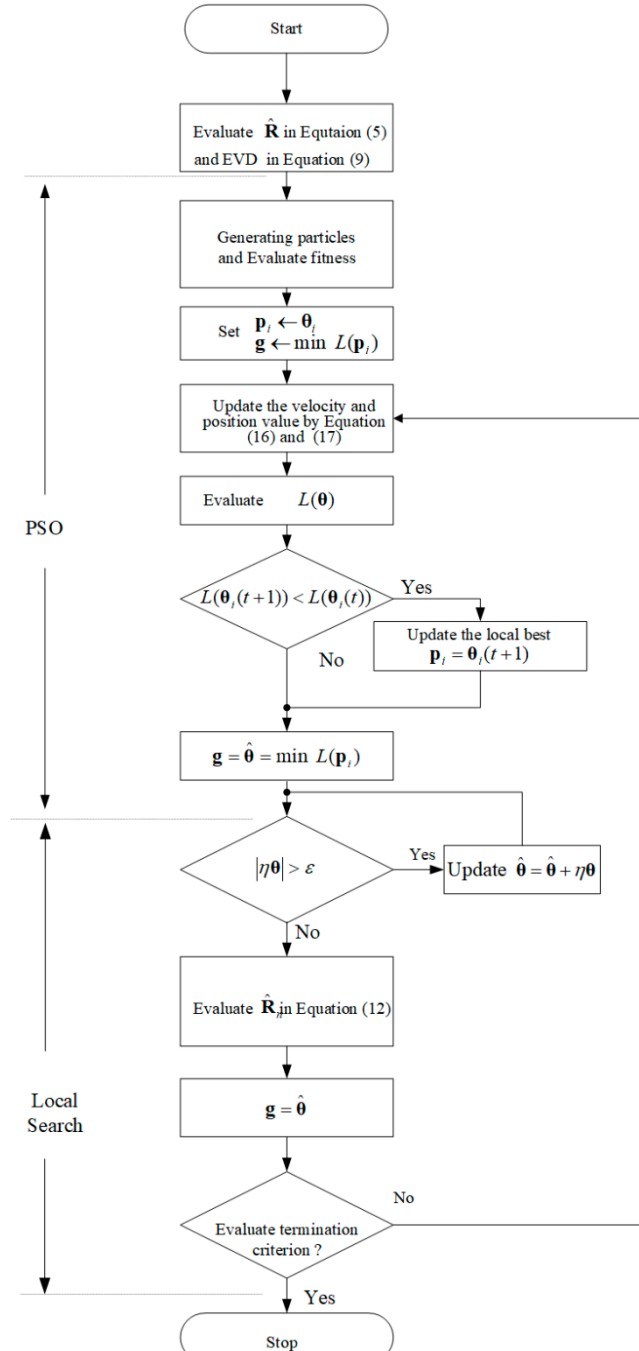

**Figure 1.** The flowchart of the proposed method.

## 4. Simulation Result

Two examples were considered in order to illustrate the practicability of using the proposed algorithm for DOA estimation in a non-uniform noise and low SNR environment. Simulation results were used to compare the performance of the proposed algorithm with the MUSIC [14], minimum variance distortionless response (MVDR) [1], and power domain ML (Power-Domain) methods [10]. The non-uniform noise (using the worst noise power ratio; WNPR) and SNR were defined using:

$$\text{WNPR} = \frac{\sigma_{\text{max}}^2}{\sigma_{\text{min}}^2},$$

$$\text{SNR} = \frac{s_p^2}{M} \sum_{i=1}^{M} \frac{1}{\sigma_i^2}, \ for \ p = 1, 2, \cdots, P, \tag{20}$$

where $\sigma^2_{\max}$ and $\sigma^2_{\min}$ are the maximum and minimum non-uniform noise variances, respectively. The simulated results were obtained by averaging 500 independent Monte Carlo (MC) runs. In the literature, a variety of statistical methods have been applied to compare performance. These include the mean absolute error (MAE) and the root-mean-squared error (RMSE), which are defined as

$$\text{MAE} = \frac{1}{500 \times P} \sum_{j=1}^{500} \sum_{p=1}^{P} \left| \hat{\theta}_{j,p} - \theta_p \right|,$$

$$\text{RMSE} = \sqrt{\frac{1}{500 \times P} \sum_{j=1}^{500} \sum_{p=1}^{P} \left( \hat{\theta}_{j,p} - \theta_p \right)^2}, \tag{21}$$

where $\hat{\theta}_{j,p}$ is the *j*th MC run for the $\theta_p$ estimate value. The search grid capacity for the spectrum scan of MUSIC [14] was set as 0.001°. The initial set of parameters of the proposed estimators were defined as

$$c_1 = c_2 = 2.05, \chi = 0.99, S = 200, T = 50, \varepsilon = 0.001, \tag{22}$$

where $c_1$, $c_2$ are acceleration coefficients, $\chi$ is the inertial weight, *S* is the size of the swarm, *T* is the number of iterations, and $\varepsilon$ is the search precision value.

The first example considered a four-element ULA with half-wavelength inter-element spacing, where the noise power was given as $\sigma^2 = \begin{bmatrix} 5 & 10 & 0.1 & 6 \end{bmatrix}^t$, WNPR = 50, and the source had the DOA $\theta_1 = 5°$. Figure 2 illustrates the RMSE values of the estimated DOA versus the number of snapshots under SNR = −15 dB. The proposed method achieved a faster convergence approach with 75 snapshots, while the other methods converged at about 150 snapshots. Figure 3 shows that the RMSE of the various estimators versus different SNRs under snapshots was 100. In Figure 3, we can see that all estimator RMSEs were near zero under high SNR, but the performance of other estimators decayed under low SNR conditions. Table 1 shows the RMSE and MAE under various SNRs. The bold text is the best estimated value under the same SNR. The proposed algorithm had RMSE and MAE values smaller than those of the other estimators, especially under low SNR environments.

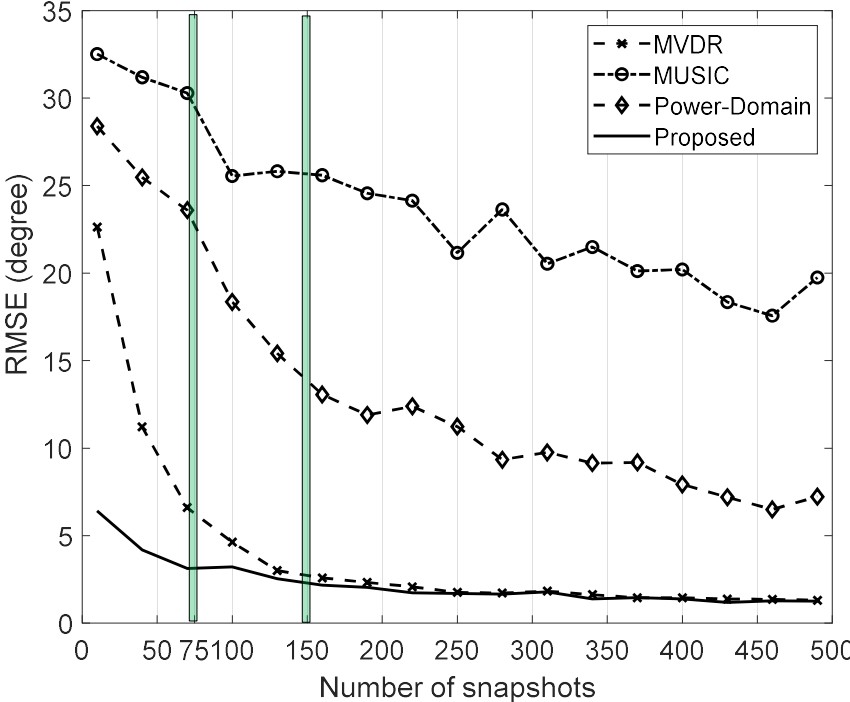

**Figure 2.** Root-mean-squared error (RMSE) versus the snapshot for the different estimators (MUSIC, MVDR, Power-Domain, and the proposed estimator) for Example 1.

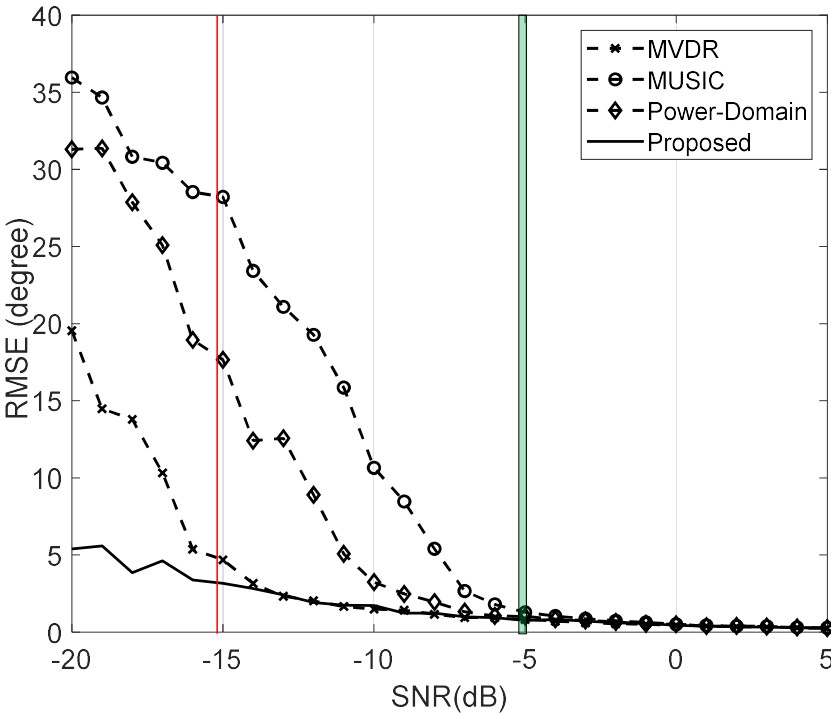

**Figure 3.** RMSE versus the signal-to-noise ratio (SNR) for the different estimators (MUSIC, MVDR, Power-Domain, and the proposed estimator) in Example 1.

**Table 1.** Direction of arrival (DOA) evaluation for the different estimators (multiple signal classification (MUSIC), minimum variance distortionless response (MVDR), power domain maximum likelihood (Power-Domain), and the proposed estimator) in Example 1.

|  |  | MVDR | | MUSIC | | Power-Domain | | Proposed Method | |
|---|---|---|---|---|---|---|---|---|---|
|  |  | RMSE | MAE | RMSE | MAE | RMSE | MAE | RMSE | MAE |
| SNR | −20 dB | 19.5392 | | 35.9564 | | 31.3039 | | 5.3792 | |
|  | −15 dB | 4.6780 | | 28.2148 | | 17.6577 | | 3.1546 | |
|  | −10 dB | 1.4788 | | 10.6452 | | 3.2314 | | 1.7144 | |
|  | −5 dB | 0.8232 | | 1.2765 | | 1.0158 | | 0.7440 | |
|  | 0 dB | 0.4282 | | 1.2765 | | 0.5046 | | 0.4368 | |
|  | 5 dB | 0.2368 | | 0.2728 | | 0.2545 | | 0.2392 | |

The second example considered an eight-element ULA and two sources with DOAs $\theta = \begin{bmatrix} -3^{\circ} & 6^{\circ} \end{bmatrix}^{t}$. The additive background noise variance was $\sigma^2 = \begin{bmatrix} 6 & 2 & 0.5 & 2.5 & 3 & 1 & 1.5 & 10 \end{bmatrix}^{t}$ and WNPR = 20. The other parameters were the same as in Example 1. Figure 4 indicates the RMSE of the various estimators versus different snapshots under SNR = 0 dB. Again, this figure indicates that the proposed estimator not only carried out faster convergence (at 50 snapshots) but also offered an improvement in DOA evaluation accuracy. Figure 5 indicates that the RMSE of the various estimators versus different SNRs under snapshots was 100. In Figure 5, we can see that the other estimators were not sensitive to various low SNRs (SNR < −5 dB), when comparing their performance with the proposed method. As hoped, the results indicate again that the DOA evaluation accuracy was improved by the proposed estimator. Table 2 gives the DOA estimates (RMSE, MAE) of the MUSIC, MVDR, and Power-Domain methods, along with those of the proposed estimator. The evaluation accuracy of the MUSIC, MVDR, and Power-Domain estimators worsened under a low SNR. Moreover, the accuracy of the proposed estimator was better than those of the other estimators under low SNR conditions.

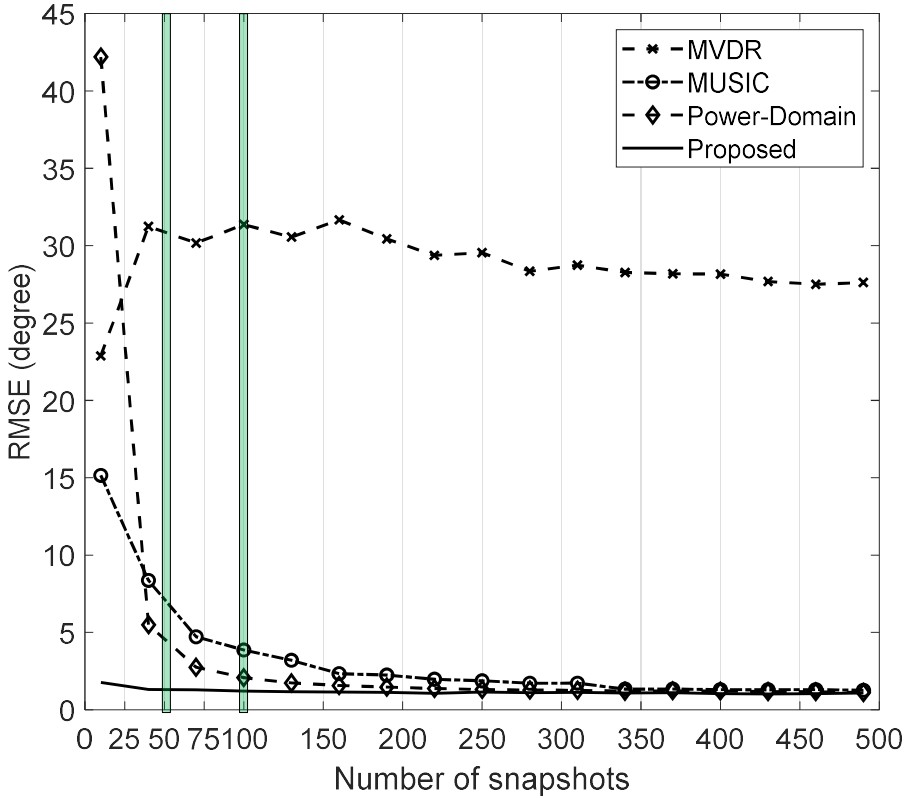

**Figure 4.** RMSE versus snapshot for the different estimators (MUSIC, MVDR, Power-Domain, and the proposed estimator) in Example 2.

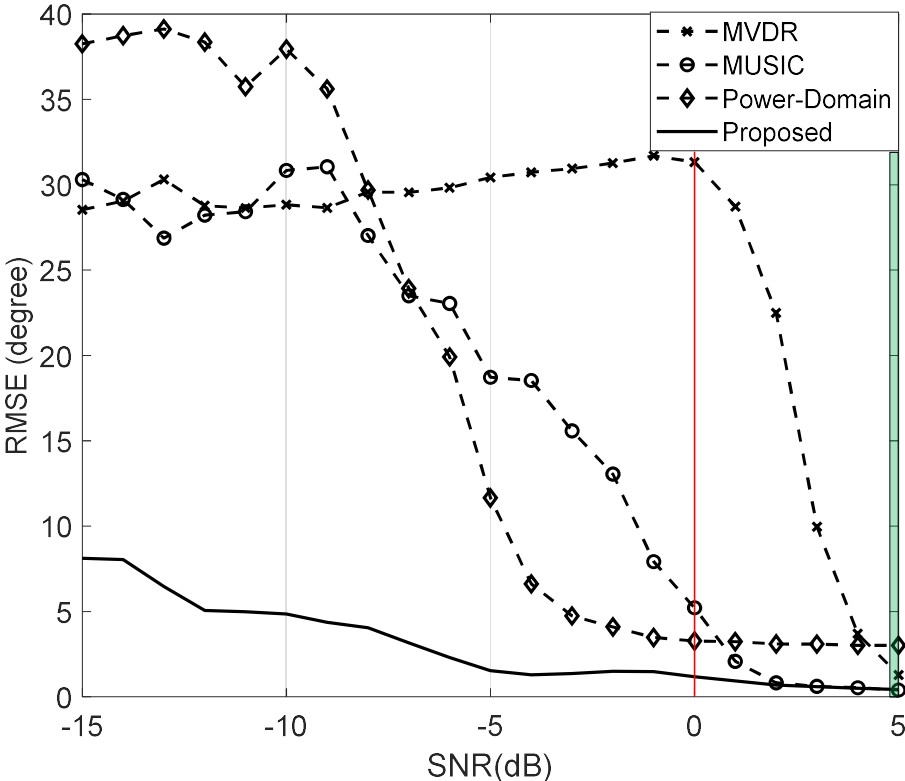

**Figure 5.** RMSE versus SNR for the different estimators (MUSIC, MVDR, Power-Domain, and the proposed estimator) in Example 2.

**Table 2.** DOA evaluation for the different estimators (MUSIC, MVDR, Power-Domain, and the proposed estimator) in Example 2.

|  |  | MVDR | | MUSIC | | Power-Domain | | Proposed Method | |
|---|---|---|---|---|---|---|---|---|---|
|  |  | RMSE | MAE | RMSE | MAE | RMSE | MAE | RMSE | MAE |
| SNR | −15 dB | 28.5391 | 22.2818 | 30.3017 | 25.5400 | 38.2552 | 30.3440 | 8.1092 | 7.3989 |
|  | −10 dB | 28.8271 | 22.4764 | 30.8397 | 24.0450 | 37.9480 | 29.2230 | 4.8597 | 4.4936 |
|  | −5 dB | 30.4322 | 23.3296 | 18.7202 | 15.3910 | 11.6566 | 9.4562 | 1.5421 | 1.4312 |
|  | 0 dB | 31.3296 | 24.2212 | 5.2164 | 4.4160 | 3.2706 | 2.9772 | 1.1865 | 1.1242 |
|  | 5 dB | 1.2848 | 1.2488 | 0.3967 | 0.3590 | 3.0139 | 2.9830 | 0.4313 | 0.3950 |

## 5. Conclusions

DOA estimation cannot be directly carried out under a non-uniform noise and low SNR environment, in which the closed form of the contained criteria is $\{\theta, \sigma^2\}$. Generally, it is necessary to process the DOA ($\theta$) and the noise variance ($\sigma^2$). In this paper, a new re-iterated process was introduced, in which the noise variance is fixed to estimate the DOA and, vice versa, the DOA is fixed to estimate the noise variance. After several iterations, the procedure converges to the nearest correct estimates of the DOA and the noise variance. The proposed solution combines the MPSO scheme, which uses the fixed noise variance to estimate the DOA through the PSO algorithm, using the best particle to estimate the noise variance. An MPSO that incorporates the re-estimation of noise variance and an iterated local search algorithm is applied in the PSO, resulting in an efficient reduction of the non-uniform noise effect under a low SNR. The iterated local search of the MPSO method exploits the characteristics of the first-order Taylor expansion $|\eta\theta|$. A small $|\eta\theta|$ value can guarantee convergent results that may be local or global while, with a large $|\eta\theta|$ value, convergence will take a longer time and may provide a value that updates the estimated deviation toward the correct value. Empirical evidence shows that the DOA estimation methods are valid in a high SNR environment, but in a low SNR and non-uniform noise environment, the performance becomes poor because the noise is confused by the source of the signal. The proposed method incorporates the re-estimation of noise variance and an iterated local search algorithm in the PSO. This method is effectively able to reduce estimation deviation and to achieve high accuracy and fast convergence in low SNR and non-uniform environments. Generally, a low SNR and non-uniform environment is caused by foul weather. This problem occurs in mountaineering, so this method provides valid DOA estimation in this environment, and can be used for positioning system issues.

**Funding:** This research received no external funding.

**Conflicts of Interest:** The author declares no conflict of interest.

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
