# Peer review of "DOA Estimation in Non-Uniform Noise Based on Subspace Maximum Likelihood Using MPSO"

_processes, doi:10.3390/pr8111429_

Round 1

Reviewer 1 Report

The paper is not intelligibile. The quality of English language and grammar is particularly low, and this impedes the understanding of its content.

I recommend revising it with the support of an English native speaker and re-submitting it for peer-review.

Reviewer 2 Report

The content of the paper is of interest to the scientific community.  The paper would benefit greatly for editing of the English.  Example:

The first part of the Abstract should probably read: "In general, the performance of DOA estimators may decay under nonuniform noise and low signal-to-interference ratio (SNR) environments. This paper proposes memetic particle swarm optimization (MPSO) combined with estimation noise variance to address this issue. The MPSO incorporates re-estimation noise variance and local search techniques applied to particle swarm optimization (PSO) resulting in efficiencies and reductions in nonuniform noise effects under low SNR. The procedure for MPSO is as follows: first, the PSO is used to estimate the signal DOA by a subspace maximum-likelihood (SML) method...."

Indicative (but not exhaustive) attention to editing is as follows:

line 23: delete "an"

line 32: MUSIC typo

line 37: "will dangerous", meaning unclear

lines 41, 62, 63, 91, 140, 183 etc: reconsider use of commas

line 163 to 170: "The introduction should briefly place the study in a broad context and highlight why it is
164 important. It should define the purpose of the work and its significance. The current state of the
165 research field should be reviewed carefully and key publications cited. Please highlight controversial
166 and diverging hypotheses when necessary. Finally, briefly mention the main aim of the work and
167 highlight the principal conclusions. As far as possible, please keep the introduction comprehensible
168 to scientists outside your particular field of research. References should be numbered in order of
169 appearance and indicated by a numeral or numerals in square brackets, e.g., [1] or [2,3], or [4–6]. See
170 the end of the document for further details on references." is from guidelines to authors and should not be in the paper proper.

Section 3 title is the same title as the title of subsection 3.2.

lines 205 to 213 content needs to be back at 3.0.

line 73: reducing

line 74: "for" not "at"

various: the use of "we" when the author is singular.

various: reconsider or delete the use of the modifiers: "almost", "rather"

various: rectify past tense and present tense confusion: "utilised/utilises", "presented/presents/presenting", "reducing/reduces/reduced", "used/uses/using", "are/is", "combined/combines"

Some suggestions attached in marked-up Work=d file with TrackChanges enabled.

Reviewer 3 Report

  1. In general, the abstract consists of "background","methods","results", and "conclusion". However, I could not find "results" and "conclusion" in the abstract.
  2. References must be numberred in order of appearance in the text accroding to MDPI form. However, this paper did not comply with the template.
  3. This paper did not follow figure, table text format. Also, I could not find text in figures because text sizes in figures were small. 
  4. Author must comply with references form according to the template.
  5. In this paper, author expressed "we" several times. However, I recognized that the author is only one.
  6. Abbreviation of the "MUSIC" was expressed as MSUIC in the introduction.
  7. Title of the Section 3 same as subtitle 3.2.
  8. Author tried to validate performance of the proposed method by using "MAE" and "RMSE". But I could not find results of "RMSE" and "MAPE" in Table 1. In paricular, author did not MAPE in this paper.
  9. I could not find abbreviation of "MVDR".
  10. Author developed the imporved MPSO based on existing research [1,9,11]. However, simulation results did not state comparision of the existing research [1,9,11] and the proposed method. 

Round 2

Reviewer 1 Report

With an adequate English language and grammar revision, the paper has a completely different shape, and can deliver all the positive hints eventually promised.

I have just a couple minor remarks:

- In the Conclusion section, I would give a take-home message to the reader, stating the contribution of the paper in terms of: i) advancement to the State-of-the-Art, ii) possible applications and usefulness in “real life”, iii) future developments of the present work.

- Table 1. “-05dB” should be “-5dB”

Reviewer 3 Report

  1. Abstract was significatly improved when comparing with old version. Howerver, author neet to note overview of the research findings and main results.
  2. In section 3, sentance (170-184) have to be arranged in introduction section. Author only need to suggest the proposed method with used materials except on identical explanantion likewise sentance (170-184).
  3. In section 3.2, author neet to arrange the proposed algorithm using box.
  4. In equation (20), (21), italic of WNPR, SNR, MAE, and RMSE were different, respectively. Author thus neet to unify them.
  5. The conclustion does not adequatedly summarize the main findings of the investigation. Only qualitative result was indicated that " Empirical evidence indicates that the proposed algorithm has an enhanced capacity to decrease the estimation bias under non-uniform noise and low SNR conditions." I suggetst that the conclusion should be revised.
